# A Comprehensive Sequencing Analysis of Testis-Born miRNAs in Immature and Mature Indigenous Wandong Cattle (*Bos taurus*)

**DOI:** 10.3390/genes13122185

**Published:** 2022-11-23

**Authors:** Hongyu Liu, Ibrar Muhammad Khan, Yong Liu, Nazir Muhammad Khan, Kaiyuan Ji, Huiqun Yin, Wenliang Wang, Xinqi Zhou, Yunhai Zhang

**Affiliations:** 1Anhui Provincial Laboratory of Local Livestock and Poultry Genetical Resource Conservation and Breeding, College of Animal Science and Technology, Anhui Agricultural University, Hefei 230036, China; 2Anhui Province Key Laboratory of Embryo Development and Reproduction Regulation, Anhui Province Key Laboratory of Environmental Hormone and Reproduction, School of Biological and Food Engineering, Fuyang Normal University, Fuyang 236037, China; 3Department of Zoology, University of Science and Technology, Bannu 28100, Pakistan; 4Reproductive Medicine Center, the 901st Hospital, Hefei 230031, China

**Keywords:** bovine, testicular growth, spermatogenesis, miRNA, total RNA-Seq

## Abstract

Micro RNAs (miRNAs) have been recognized as important regulators that are indispensable for testicular development and spermatogenesis. miRNAs are endogenous transcriptomic elements and mainly regulate the gene expression at post-transcriptional levels; however, the key role of miRNA in bovine testicular growth is not clearly understood. Thus, supposing to unveil the transcriptomics expression changes in the developmental processes of bovine testes, we selected three immature calves and three sexually mature bulls of the local Wandong breed for testicular-tissue sample collection. The cDNA libraries of experimental animals were established for RNA-sequencing analysis. We detected the miRNA expression in testes by using high-throughput sequencing technology, and bioinformatics analysis followed. The differentially expressed (DE) data showed that 151 miRNAs linked genes were significantly DE between immature and mature bull testes. Further, in detail, 64 were significantly up-regulated and 87 were down-regulated in the immature vs. mature testes (*p*-value < 0.05). Pathway analyses for miRNA-linked genes were performed and identified *JAG2, BCL6, CFAP157, PHC2, TYRO3, SEPTIN6*, and *BSP3*; these genes were involved in biological pathways such as TNF signaling, T cell receptor, PI3KAkt signaling, and functions affecting testes development and spermatogenesis. The DE miRNAs including MIR425, MIR98, MIR34C, MIR184, MIR18A, MIR136, MIR15A, MIR1388 and MIR210 were associated with cattle-bull sexual maturation and sperm production. RT-qPCR validation analysis showed a consistent correlation to the sequencing data findings. The current study provides a good framework for understanding the mechanism of miRNAs in the development of testes and spermatogenesis.

## 1. Introduction

The formation of mammalian testes is a complicated process comprising numerous molecular and cellular procedures that culminate in the progressive differentiation of many different cell types, each with its unique set of activities. Spermatogenesis involves the endocrine and paracrine pathways, as well as the meiotic cell division, which eventually produces haploid spermatozoa [1,2]. The reproductive performance of cattle bulls is an important factor in the biological breeding and economic efficiency of cattle production, whereas poor reproductive performance has also raised important issues in the dairy and beef industries worldwide [3,4]. However, the reproductive improvement in local-breed bulls, via an emphasis on their hidden potential genetic resources that regulate the testis development and spermatogenesis at transcriptional and post-transcriptional levels, remains unmapped.

The identification of genes involving testicular growth and spermatogenesis is important to determine valuable insights into the mechanism of functional sexual maturation [5]. MiRNA is a type of non-coding RNA (ncRNA) with a length of about 22 nucleotides, and it inhibits target gene translation by binding to the complementary sequences in the 3UTR [6,7]. Additionally, miRNAs regulate the target gene expressions both at the post-transcriptional and the transcriptional levels by the RNA–RNA interactions [8]. A significant number of miRNAs have been discovered to play key roles in different physiological processes including cell growth, zygote development, spermatozoa morphology and other kinetic parameters [9,10]. MiRNA play an imperative role in mammalian reproduction, including in gametogenesis [11], fertilization [12], zygotic genome activation, early development [13], early implantation [14], germ cell specification and modification [15], sexual differentiation [16], and pregnancy [17]. Herein, it is concluded from the rational studies that miRNA significantly mediated the regulation of bovine reproduction, including germ cell biogenesis, reproductive organs functioning, and growth.

The proportional expression profiling of miRNAs in the neonatal, young adult, and aged human epididymides revealed that neonatal epididymis expressed 127 miRNAs pre-dominantly, whereas only a few miRNAs were abundantly expressed in the adult and aged epididymides, respectively [18]. MiRNA has been identified in the ovary and testis of porcine, whereas the co-expression patterns of X-linked miRNAs were only found in adult gonads [19]. The miRNA expression variations in sexually immature (60-day) and mature (180-day) pig testes were also studied, and showed that miRNAs play a significant role in controlling spermatogenesis [20]. MiR-22 blocks estrogen receptor 1 (ESR1), which mediates estrogen signaling in the ovine fetal testes, in resulting male gonadal establishment [21]. Specifically, in testicular tissue, the expression of, miR-202-3p, miR-202-5p, miR-140-3p, and miR-140-5p is up-regulated, indicating that they may play a role in testicular growth and Leydig cell formation [22]. Other unique miRNAs, such as miR-878-5p and miR-485, have a time-dependent richness in the ovary and testes of mice, suggesting that they are essential for sex determination [23]. Numerous miRNAs have been found in the male reproduction of bovine species, where the DE pattern in the mechanism of spermatogenic arrest in cattle, yaks, and their interspecies is being researched [24,25,26]. The expression of miRNAs in the bovine spermatozoa as well as the process of spermatogenic arrest in cattle and yak has been identified in several studies. However, the expression of miRNAs and their regulation tools during the developmental stages of bovine testes remains uncertain.

This research is based on total RNA sequencing and computational target prediction to identify miRNAs that may influence bovine reproduction. Furthermore, a significant number of novel and candidate genes were discovered in Wandong bull testes that control the reproduction traits, including testicular growth and spermatogenesis.

## 2. Materials and Methods

### 2.1. Ethical Approval

The College of Animal Science and Technology, Anhui Agricultural University and Animal Care Unit, approved the collection of experimental animals under strict ethical procedures (SYXK 2016-007). Necessary measures were taken to keep the experimental animals as pain-free as possible.

### 2.2. Experimental Animals, Sample Collection and Histological Assessment

A Chinese local breed known as Wandong cattle in Anhui Province was selected for the current study. Six experimental animals were slaughtered at two different developmental stages: the immature group was 3 ± 0.24 months and the sexually mature was 3 ± 0.014 years old [27]. Healthy animals were selected, and the matured group was sexually active. Soon after slaughtering, the testes were removed and crosscut into the size of 5 mm × 5 mm and frozen in liquid nitrogen. All pairs of testes were processed by incising the scrotum medially and uncovering the right and left testicles within the tunica vaginalis. After that, the tissues were preserved in a 4% formaldehyde solution for the histological assessment [28], whereas some tissue samples were stored at −80 °C for RNA sequencing.

### 2.3. Small-RNA Library Construction for Illumina

The small-RNA library was generated using a total of 10 μg RNA per sample as input content. For the small-RNA indexed libraries, NEBNext^®^Multiplex small-RNA Library Prep Set for Illumina^®^ (NEB, County Rd, Ipswich, MA, USA) was used while following the manufacturer’s guidelines. Each sample’s attribute sequences were modified with the index codes. It was recommended that the NEB 3′SR Adaptor be used to ligate the 3′ end of small RNAs. The SR RT Primer hybridized to certain 3′SR Adaptors that were left free after the 3′ ligation reaction, transforming all single-stranded DNA adaptors to double-stranded DNA molecules. This key step prevents the formation of adaptor-dimer, as dsDNA is not reacted with ligation mediated by T4 RNA Ligase 1 and therefore not linked to the 5′SR Adaptor in the subsequent ligation stage. Then, using the M-MuLV Reverse Transcriptase (RNase H), we synthesized the first strand of cDNA. PCR amplification was performed using LongAmp Taq 2X Master Mix, SR Primer for Illumina and index (X) primer. At last, library quality was assessed on the Agilent Bioanalyzer 2100 system using DNA High Sensitivity Chips and the flow chart shown below (Appendix A).

### 2.4. Illumina Sequencing and Bioinformatics Analysis

According to the manufacturer’s instructions, the index-coded samples were clustered using the Illumina supporting core (cBot Cluster Generation System using TruSeq SR Cluster Kit v3-cBot-HS). After clustering, the library was sequenced on the Novogene Illumina Hiseq 2500 platform (Beijing, China), and 50bp single-end reads were generated [29]. In the beginning, raw fastq data were processed using custom Perl and Python scripts. The reads consisting of ploy-N, with 5′ adapter contaminants, without 3′ adapter, or the insert tag, which contains ploy A, T, C, or G, and the poor-quality reads were removed to obtain clean reads. The Phred value, Q20, Q30, and GC-content were also calculated, and the Q20 score was selected to choose clean reads for downstream analyses. The snRNA, snoRNA, tRNA, and rRNA free reads of each calf and bull sample were charted to reference genome *Bos taurus*-UMD 3.1.1 by TopHat2 (v. 2.0.3.12), as described in reference [30].

### 2.5. Known and Novel miRNAs Prediction

The database of miRBase v20.0 and the modified mirdeep2 software was used to identify known miRNAs [31], srna-tools-cli were was used to obtain the potential miRNA, and custom scripts were used for drawing the secondary structures. The properties of the hairpin structure of the miRNA precursor could be used to predict novel miRNA. The miREvo, 1.2 and mirdeep2.0.1.2 tools were used to predict novel miRNAs, which were foreseen based on the following properties: we looked at the secondary structure, Dicer cleavage site and lowest free energy of the un-annotated sRNA tags [32]. The schematic diagram is shown in Appendix A.

### 2.6. Differentially Expressed miRNA Analysis

A Venn diagram was constructed using Microsoft Excel 2016 to visualize the DE miRNA changes by the two stages. A STEM program was used and identified the expression profiles of miRNAs (http://www.cs.cmu.edu/-jernst/st/, accessed on 19 November 2019). The DE miRNAs in two groups of bovine testes were categorized into three expression patterns. Hierarchical cluster analysis was performed with Cluster 3.0 and TreeView1.6 programs (http://rana.lbl.gov/eisen, accessed on 19 November 2019). The miRNAs gene expression level was assessed with the TPM (transcript per million) value and using the DESeq R package (1.8.3) [33]. A transcript-level analysis of DE miRNAs was performed, and functional genes related to the developmental groups were discovered. The Benjamini and Hochberg method was used to change the *p*-values, and a corrected *p*-value of 0.05 was chosen as the threshold for substantially different expressions.

### 2.7. GO and KEGG Enrichment Analysis

The miRanda tool [34] was used to predict miRNA target genes, and KOBAS software was used to investigate the statistical enrichment of KEGG pathways. The GO enrichment summary and classification provides significant DE miRNAs target genes that participated in various biological functions. During the GO enrichment analysis, the GO-seq-based Wallenius non-central hypergeometric distribution was used to adjust gene length bias [35]. All of the pathway source genes and background genes were mapped to GO terms in the database (http://www.geneontology.org/, accessed on 19 November 2019). The hypergeometric test was used to identify the GO terms that were significantly enriched by DE genes. A KEEG pathways-based study further elaborated the explanation of source genes and their biological function (http://www.genome.jp, accessed on 19 November 2019). KOBAS 3.0 software was suggested to test enriched DE source genes in KEGG pathways [36].

### 2.8. Integrated Analysis among miRNAs–mRNAs and DE genes

For the miRNA-target gene network, BLASTN (https://blast.ncbi.nlm.nih.gov, accessed on 25 December 2019) was used to detect and eliminate pre-miRNAs based on substantial amounts of similarity. Then, MiRanda 0.8.4 [34] was used to predict the target relationships with miRNAs, which needed an alignment score of N ≥ 140, and required energy less than −10 kcal/mol. Based on typical miRNA-binding sites, additional analyses of miRNAs–mRNAs and gene pairs were implemented. The miRNA–mRNA and DE genes interaction map were developed and visualized using Cytoscape v3.7.2 (https://cytoscape.org, accessed on 25 April 2020) [37].

### 2.9. Real Time Quantitative PCR

The Trizol reagent (Life Technologies, 182805, Waltham, MA, USA) was used to extract RNA from testes, and the amount was determined using a Nanodrop device. The high-quality RNAs were then used to create cDNA and we employed a QuantiTect Reverse Transcription Kit (Qiagen, 205311, Stockach, Germany) as per the manufacturer’s directions. The expression levels of nine miRNAs were detected using q-PCR and using a miRcutePlusmiRNA q-PCR Kit (Tiangen Biotech Co., Ltd., Beijing, China) and SuperReal PreMix Plus (SYBR Green). The miRNA sequences were obtained as forwarding PCR primers from the NCBI database, and the primer sequences are described in Table 1. As usual in our study, we also used stem-loop RT primers, a universal reverse primer (fix sequence), and specific forward primers to amplify miRNA sequences. The primers were manufactured by the Sangon biotech company (Shanghai, China). The following are the thermal cycling parameters used for q-PCR miRNAs amplification: The pre-denaturation cycle was fixed at 95 °C for 10 min, followed by 45 amplification cycles of denaturation at 95 °C for 15 s; the annealing was fixed at 60 °C for 10 s; and an extension at 72 °C for 20 s. To normalize the relative miRNA expression, the 2^−ΔΔCt^ method was used, with the U6 small nuclear RNA serving as an internal standard. The RTq-PCR was performed in three biological replicates to minimize the chance of error when conducting the trials. The Cq values were collected and shifted to a Microsoft Excel spreadsheet for use of the 2^−ΔΔCt^ process of relative quantification analysis [38].

### 2.10. Statistical Analysis

The miRNAs were examined using the Student’s *t*-test (SPSS 17.0), and the results were given as (mean ± SEM). Normal data distribution was found between the two experimental groups and homogenous variances were measured and suggested that available data are normal and homogeneous. Mean values were considered to be significantly different, at ** p* < 0.05 and *** p* < 0.01.

## 3. Results

### 3.1. Histomorphology of Testis

The histomorphology study of testes showed a significant difference between the immature and sexually mature growth stages of bulls. The diameter of seminiferous tubules in the testicular tissue of the mature group compared to the immature was considerably large when observed at 100× microscopic magnification. A very comparable condition was revealed in the interstitial connective tissue of testes in sexually mature bulls and calves, as highlighted in Figure 1a,c. At 400×, we delved into more detail and discovered that mature testes had more advanced stages of spermatozoa than immature testes. Sertoli cells were substantially larger and more numerous in mature testes and were discovered particularly close to the basement line of seminiferous tubules [39], as shown in Figure 1b,d.

### 3.2. Transcriptomics Summary of Sequencing Data

Six cDNA libraries were assembled for the experimental calves and bulls, to study the functional role of miRNAs in testicular growth and spermatogenesis. We used RNA-Seq data libraries of testicular tissue and created the small-RNAs profile. The total average raw reads of Seq-data were acquired (2015,2658, 100.00%) after sequencing through the Illumina Hi-Seq 2500 Platform. Raw reads were filtered and classified into different segments, including clean reads (20,838,854.33, 98.38%), containing N reads (0, 0.00%), low quality (89,245.33, 0.41%), adapter contamination (3380, 0.01%), and reads with polynucleotide bases (115,334, 0.54%), for mature bulls. A similar classification was conducted for immature calves, including the clean reads (18,848,640, 98.48%), containing N reads (0, 0.00%), low quality reads (80,597, 0.40%), adapter contamination (3670.33, 0.02%) and with polynucleotide bases (23,906.33, 0.12%). The total GC contents were calculated for raw reads of each sample (48.32 to 48.99%) (Appendix A). A minimum Phred quality score of Q20 was considered for filtering the reads with low-quality bases. All the details are shown in Table 2.

### 3.3. Sequencing Data Quality

The sequencing error rate distribution test is used to assess the % of samples sequenced incorrectly, and the sequencing error rate distribution check can indicate the quality of sequencing data. The first few bases have a high sequencing error rate due to the position, because the reverse transcription requires random primers in the process of RNA-seq library construction (Figure 2a,b). The obtained clean reads of each sample were screened for s-RNA subsequent analysis, within a certain length range. Generally, the length interval of sRNA for an animal is 18 to 35 nt, and different peaks of length distribution can help to determine the type of sRNA, such as miRNA, which is concentrated in 21~22 nt, and siRNA in 24 nt. The length distribution statistics of s-RNAs are presented in Figure 2c,d.

### 3.4. Analysis of Known and Novel miRNAs

In total, 754 known miRNAs were identified in bull vs. calf testes samples, with 98 of them being novel miRNAs. The Dicer cleaved the double-stranded RNAs or precursor miRNAs molecules and transferred them into the developed molecules. The first base of a mature known miRNA sequence is strongly biased because of the restriction of site specificity. As a result, we investigated the frequency distribution of the first base of miRNAs at various lengths and positions. We noticed that 5′ prime has a strong frequency distribution of uracil (U) base at 18 to 24 lengths (nt) while resistance to G base is shown Figure 3a,b. However, the miRNAs nucleotide bias at each position showed resistance to U base at the second, third and fourth positions, and exhibited strong frequency to G base at the 5′ ends. While at base 8, the strong preference for A was placed Figure 3c,d. We also identified the novel miRNAs in the testes samples, and the analysis of novel miRNAs at the first nucleotide bias showed a dominant bias to U at the first nucleotide, particularly the miRNAs with a length of 19–24 nucleotides (Figure 4a,b). However, results of novel miRNAs bias at each position showed that the most obvious bias to U was found at the 1st, 4th, 6th, 11th, 15th, 19th, and 22nd positions of nucleotides (Figure 4c,d).

### 3.5. Characterization of miRNAs

The total reads of miRNAs are mapped and identified based on the position and direction of their genomic regions. The total reads of miRNAs were annotated with a stringent pipeline, and we identified the different sub-classes in testicular tissues, as shown in Figure 5a,b. The chromosomal localization of testis-born miRNAs was performed and total sRNAs reads of samples were compared to the total chromosomal numbers on the genome. The distribution of reads on the chromosome was checked using Circos chart, and we identified the loci that expressed these transcripts (Figure 5c,d). The expression pattern of sRNAs was determined using cuff-diff and ballgown tools after the quantification procedure. The sRNA expression levels were estimated by transcript per million (TPM), and the average value of transcripts expression is higher in immature testes compared to the mature group, as shown in Figure 5e. The TPM value was used to standardize the expression levels of known and novel miRNAs in each sample. The miRNAs expression was measured by using the TPM density distribution plot, and as a whole, we examined the gene expression pattern of the total samples (Figure 5f).

### 3.6. miRNAs Expression Profiling of Bovine Testis in Different Developmental Stages

To find out the known and novel bovine miRNAs, we compared the clean reads of each seq-library with the reference bovine miRNA precursors in miRBase 20.0. The total numbers of mapped known and novel mature miRNAs along with hairpin were predicted in the immature and mature testis samples. There are 756 transcripts of known miRNA, and 98 novels miRNA were spotted in six libraries, as shown in Figure 6a,b. The Venn diagram indicated that immature and mature groups shared 602 miRNAs whereas 98 were only expressed in calves, and 54 miRNAs were expressed in the sexually mature bull group, as shown in Figure 6c. We further recorded the differentially expressed (DE) miRNAs between calf vs. bull testes by using the ballgown tool. The significant level of miRNAs in the testes sample was calculated and taken into account as the parameter of significance, whereas the log2 fold change was considered higher than two or equal, also *p-adjusted* < 0.05. As a result, 151 miRNAs linked genes were significantly DE between immature and mature bovine testes. Further, in detail, we identified that 64 were significantly up-regulated and 87 were down-regulated in the calf vs. bull testes (*p*-value < 0.05), as shown in Figure 6d. Hierarchical clustering displays the miRNAs between libraries that revealed differences in miRNA expression as per the different stages of testis development. The genes on the left side were grouped because of their comparable expression (fold change > 2, *p* < 0.05). Columns were used to exhibit calves and bulls, and the expression gradually increased from blue to red (Figure 6e).

### 3.7. The miRNAs Targets, GO and KEGG Pathways Analysis

The target genes of all 854 known and novel miRNAs were predicted with TargetScan and miRanda. We found 8032 mRNAs and the target genes. Since miRNAs have to control the mRNA after transcription in pattern as down-regulated mRNAs and up-regulated miRNAs, up-regulated mRNAs and down-regulated miRNAs were studied separately in the age groups. The targets of miRNAs were analyzed by gene ontology and further specified in GO terms. The GO and KEGG pathway analysis was used to look into the possible roles of the DE genes in testicular development. A total of 7140 GO terms were confirmed by the GO analysis, and among these, 667 GO terms participated in cellular components, 5327 description terms belonged to the biological processes and 1148 were significantly (corrected *p*-value < 0.05) enriched in molecular functions. The DE miRNAs were enriched in bovine reproductive-relevant GO terms, including the formation of primary germ layer, cellular component organization, cell parts, intracellular parts, and organelle membrane (Figure 7a). As for the statistic pathways enrichment analysis, DE miRNAs were enriched in the top-20 KEGG rich-factors pathways such as TNF signaling pathway, PI3K − Akt signaling pathway, cell adhesion molecules (CAMs), adherens junctions, and glycine, serine, and threonine metabolism cell adhesion (Figure 7b). A total of 151 DE genes (64 up-regulated and 87 down-regulated) which are regulated by miRNAs, were enriched in GO terms relevant to bull testes development and spermatogenesis descriptions. *JAG2*, *BCL6*, *CFAP157*, *PHC2*, *TYRO3*, *SEPTIN6*, and *BSP3* are the candidate genes closely linked to spermatogenesis. Genes including *NR5A1*, *CGA*, *NDUFS6*, *RXRA*, *TGFBR1*, *TRIM28*, and *GJB3* were found to participate in reproductive system development. The source genes found in the signaling pathways, such as the transcriptional mis-regulation in cancer, TNF signaling pathway, T cell receptor signaling pathway, PI3KAkt signaling pathway, and lysine degradation, were evaluated in the top-20 statistically enriched pathways (*p* < 0.05) (Figure 7c).

### 3.8. Validation of DE miRNAs Seq-Results by RT-qPCR

To confirm the DE miRNAs in calf and bull testes, we selected nine DE miRNAs (MIR425, MIR98, MIR34C, MIR184, MIR18A, MIR136, MIR15A, MIR1388 and MIR210) to endorse the expression patterns by RT-qPCR. Their expression pattern was examined in immature and mature groups. The obtained results showed that the RNA-seq data were consistent and reliable, with specificity to the developmental phases of the bull testis (Figure 8a–i).

### 3.9. MiRNAs–mRNAs and Link Genes Interaction Analysis

To illustrate the networking analysis of miRNAs in immature and mature testes of cattle bulls, we used algorithm miRanda model and predicted the genomics targets relationship among the miRNAs, mRNAs and the DE genes. We identified 8032 novel miRNAs and their targets mRNAs and genes (calves vs. bulls), and the interaction is shown in Figure 9.

### 3.10. Functional Differentially Expressed Genes (DEGs)

DEGs in the networks were analyzed using GO and KEGG-enriched pathways. However, the DEGs from the calves and bull groups shared some suggestively enriched GO terms. For example, sexual reproduction, male gonad development, germ cell development, germ cell migration, spermatid development, and the sperm part are the GO terms and most relevant to bovine reproduction, many DE genes corresponding to the suggested GO terms are significantly enriched in signaling pathways (Table 3).

## 4. Discussion

In this developmental study, we selected two different age groups of bulls (Bos taurus) and analyzed the expression profiling of small non-coding RNAs and their potential role in testes development and spermatogenesis. The conducted work will help to understand the molecular regulating mechanism in testis development. MiRNAs were chosen as a subject because recent research suggests that miRNAs can prevent protein expression and degrade mRNA expression in germ cell proliferation and development through post-transcriptional mechanisms [40,41]. MiRNAs are identified as significant biomarkers in spermatogenesis and testis development in several studies, and they are necessary for primordial germ cell growth and spermatogenesis [11]. Testis development and spermatogenesis are the important parameters that influence bull reproductive performance. Various functional genes are mainly involved in testis development and spermatogenesis [42,43].

The cDNA libraries of testes tissues were constructed for two developmental stages, which highlight 98.38% clean reads, 48.99% GC contents and Q20 of 97%. The miRNAs mapping, annotation and chromosomal distribution were performed, whereas the expression level of novel and known miRNAs were shown by the TPM value. Very similar patterns were adopted for the identification and characterization of miRNAs in the study of embryonic mortality in pigs [44]; in descended testes and undescended testes in horses [45]; in the immature and mature testes of the Mongolian horse [46]; and in 2-, 6- and 12-month-old small-tail Han-sheep testes [47].

We compared the miRNA profiles in immature and sexually mature testes to identify miRNAs that may control the testes’ development and spermatogenesis. A total of 754 mature miRNAs were expressed, with 602 of these being co-expressed in both the calf and bull libraries; whereas 98 were expressed in calves and 54 miRNAs were expressed in sexually mature bulls. Furthermore, 151 DE miRNAs linked genes were identified in calf vs. bull, whereas 64 genes were significantly up-regulated and 87 down-regulated. A similar comparison of DE 10716 mRNAs, 67 miRNAs and 16953 piRNAs was performed in the pig breeds at different age stages, whereas 14 miRNAs and 18 targeted link genes were supposed to have a relationship with the development of the testis [48]. In the testicular and ovarian tissues, a total of 246 known miRNAs are co-expressed; where 21 miRNAs testis-specific and 9 ovary-specific were described by [49]. Noveski [50] also looked into the role of miRNAs in spermatogenesis in hypospermia patients, indicating that miRNAs can influence normal seminiferous activity.

To methodically explain the biological functions of miRNAs and link target genes in bull testes, GO and KEGG studies were conducted for target genes of DE miRNAs. Usually, miRNAs play an important role in the regulation of target genes by adjusting the production of encoded proteins post-transcriptionally [51]. The DE genes such as *JAG2, BCL6, CFAP157, PHC2, TYRO3, SEPTIN6, NR5A1* and *BSP3* were found in our study, which were enriched in the GO terms relevant to testes development and spermatogenesis descriptions. Among these predicted target genes, *BCL6* is involved in rooster testicular development and reproduction [52]. The *TYRO3* gene is normally expressed by Sertoli cells during the postnatal development of mice, and male mice lacking in this produce no mature spermatozoa [53]. The structural variants of *NR5A1* genes are associated with the development of the sex-disordered phenotype in Yorkshire terrier dogs [54]. The transcriptional mis-regulation in cancer, TNF, T cell receptor, PI3KAkt, and lysine degradation signaling pathways were evaluated for DE source genes and found to be associated with cattle-bull reproduction. During the spermatogenesis, bta-miR-2387 target gene PTK2 regulates the PI3K-Akt signaling pathway as well as the chemokine signaling pathway, allowing spermatids to travel through the epithelium and pre-leptotene spermatocytes to cross the blood–testis barrier [55]. The DE miRNAs (MIR425, MIR98, MIR34C, MIR184, MIR18A, MIR136, MIR15A, MIR1388 and MIR210) in calf and bull testes were selected for validation purposes. All the selected miRNAs are linked to bovine reproduction. Rams were kept in a state of under-nutrition and checked for sperm quality, as under-feeding increases the apoptotic germ cells, and also increasing the expression of apoptosis-related miRNAs such as miRNA-98 was identified to target the apoptotic genes (*TP53*, *CASP3*, *FASL*), [56]. MicroRNAs (MIR34C) are active post-transcriptional gene silencing effectors, and new research indicates that the miR-34 family plays a part in bovine spermatogenesis and early embryogenesis [57]. Results demonstrated that miR-34c over-expression promoted mGSCs apoptosis and reduced their proliferation in dairy male goats [58]. Wu [59] discovered that miR-184 levels increased during mouse postnatal testis formation and that miR-184 overexpression facilitated germ cell line proliferation. A previous study suggested that certain members of the miR-17-92 cluster may play crucial functions in spermatogenesis, and the disruption of targets of miR-17, miR-18a and miR-20a could result in extreme testicular atrophy, hollow seminiferous tubules and poor sperm production in adult mice [60]. In teleost animals, gene ontology research revealed that miR-1388 target genes were highly involved in cell–cell adhesion and were located in Sertoli cells and spermatocytes through in situ hybridization [61].

## 5. Conclusions

The findings raveled that miRNAs may play an important role in bovine testicular development. We thoroughly analyzed the GO descriptions and found the source genes enriched in gonad development, spermatogenesis, spermatid development, spermatid differentiation and the reproductive process. We identified several DE target genes and miRNAs that affect testicular growth and spermatogenesis. The study of differentially expressed genes shows a regulatory molecular pattern that influences testicular growth. The discovery of functional miRNAs in the bovine genome may contribute to the biomarkers catalog for the selection and propagation of local breeding bulls.

## Figures and Tables

**Figure 1 genes-13-02185-f001:**
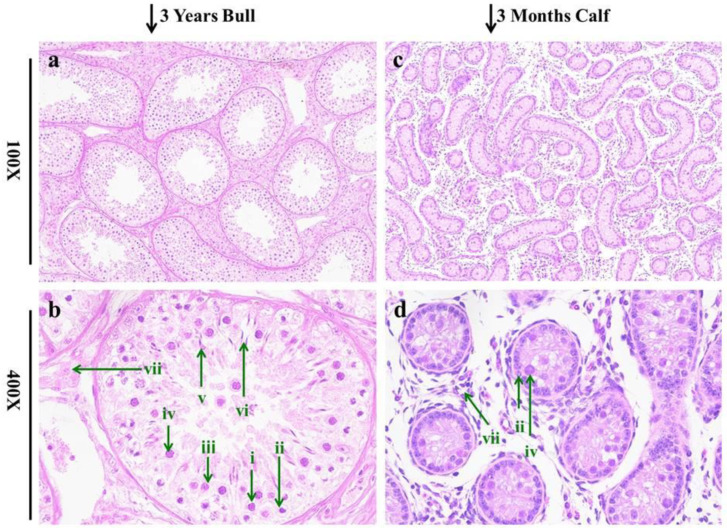
The histomorphology of bull and calf testicular tissues were studied under a microscope at 100× and 400× magnifications. Sections (**a**,**b**) characterize the morphology of 3-year-old bull testes, whereas sections (**c**,**d**) indicate the morphology of 3-month-old calf testes. The green arrows show the biogenesis of germ cells. i: spermatogonia, ii: Sertoli cells, iii: spermatocytes, iv: round spermatids, v: elongated spermatids, vi: mature spermatozoa. vii: Leydig cells.

**Figure 2 genes-13-02185-f002:**
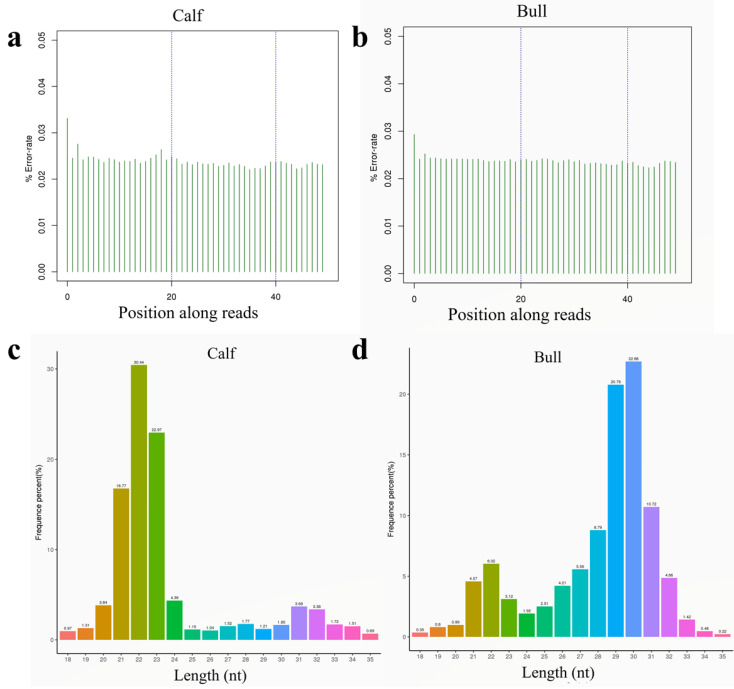
The % error rate and sRNAs length screening was shown in testicular tissue. (**a**) The abscissa represents the base position of reads, and the ordinate shows the error rate of a single base in calf. (**b**) Sequencing error rate distribution in the mature bull. (**c**) The abscissa showed the length of reads, and the ordinate is the proportion of reads in the calf. (**d**) Total s-RNA fragment length distribution statistics in the mature bull.

**Figure 3 genes-13-02185-f003:**
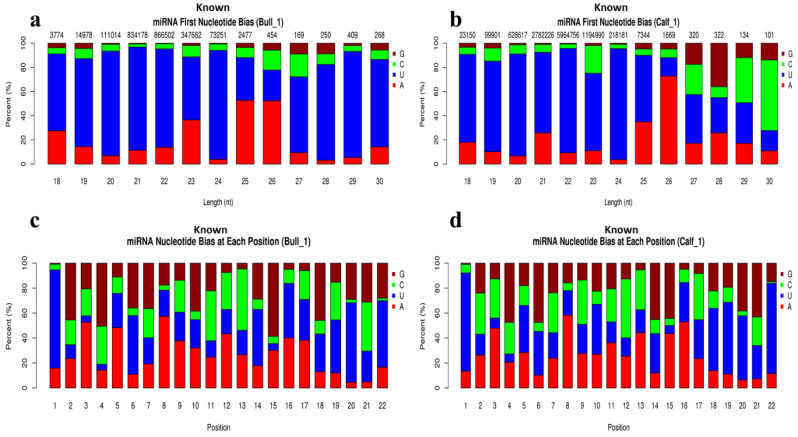
The first nucleotide bias, and bias at each position of known miRNAs were found in bull and calf testes. (**a**) The X-axis represents the length of known miRNAs and Y-axis shows the percent ratio of G/C/U and A bases. (**b**) Shown the analysis of first base preference at calf testes sample. (**c**) Exhibits the nucleotide bias at each position in bull testes and, (**d**) The abscissa shows the position of nucleotide in miRNAs, and ordinate presents responding percentages ratio of G/C/U/A bases each nucleotide in calf testes.

**Figure 4 genes-13-02185-f004:**
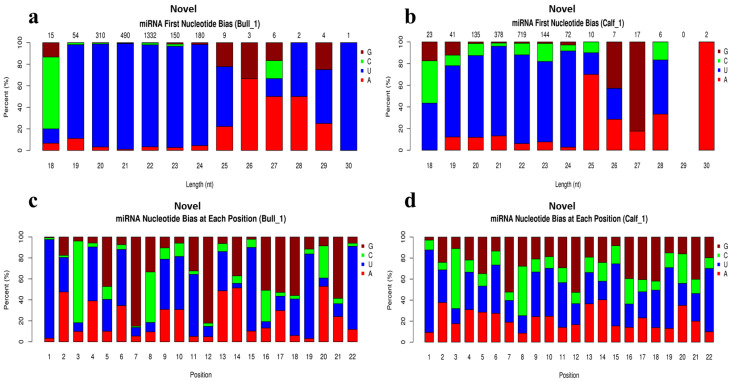
Novel miRNAs first nucleotide bias and bias at each position were investigated in bull and calf testes. (**a**) The X-axis represents the length of novel miRNAs and Y-axis shows the percent ratio of G/C/U and A bases. The (**b**) showed the analysis of first base preference in calf testes sample. (**c**) Exhibits the nucleotide bias at each position in bull testes. (**d**) The abscissa of the findings shows position of nucleotide in novel miRNAs; and ordinate shows the responding percentages ratio of G/C/U/A bases for each nucleotide in calf.

**Figure 5 genes-13-02185-f005:**
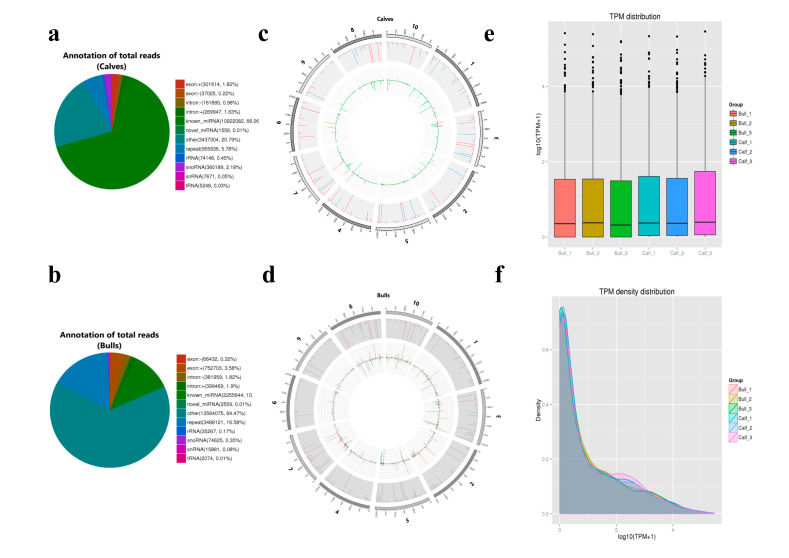
The general features and characteristics of miRNAs in bull and calf testicular tissues were tested. (**a**,**b**) The schemes represent the different annotated sub-classes of miRNA. (**c**,**d**) The miRNAs are distributed at chromosomal sets in the bovine genomics. The outer-most ring showed the chromosomes selection. In the middle, the gray background area showed the distribution of reads, whereas red mapping highlighted the positive chain and blue mapping the negative chain. The innermost circle presents the reads comparison among the chromosomes. (**e**) A box graph was used to depict the expression levels of transcripts in various bull and calf samples. (**f**) The X-axis shows the log10 (TPM + 1) value of miRNA, and the ordinate indicates the density of the corresponding log10 (TPM + 1).

**Figure 6 genes-13-02185-f006:**
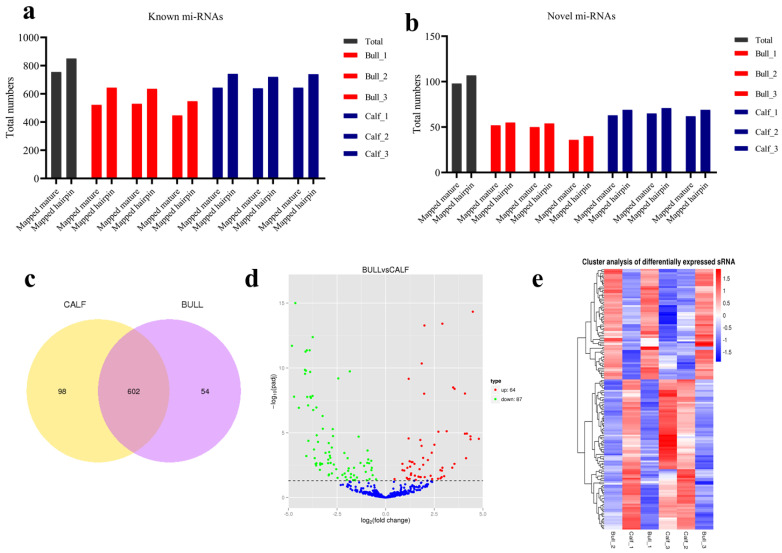
MiRNA expression patterns in bull and calf testes. (**a**) The known mapped miRNA identified in testes sample. (**b**) Novel mapped miRNAs were identified in corresponding samples. (**c**) Venn diagram shows the miRNA differential expression. The numbers in both left and right circles represent the total number of miRNAs, which are differentially expressed in each group; the overlapping part of the circles showed the total number of miRNAs expressed in both calf and bull groups. (**d**) The volcanic plot visualized the overall distribution of miRNAs and significant DE link genes. (**e**) Hierarchical diagram of differentially expressed miRNA. Red represents highly expressed miRNAs, while blue represents low–expressed miRNAs.

**Figure 7 genes-13-02185-f007:**
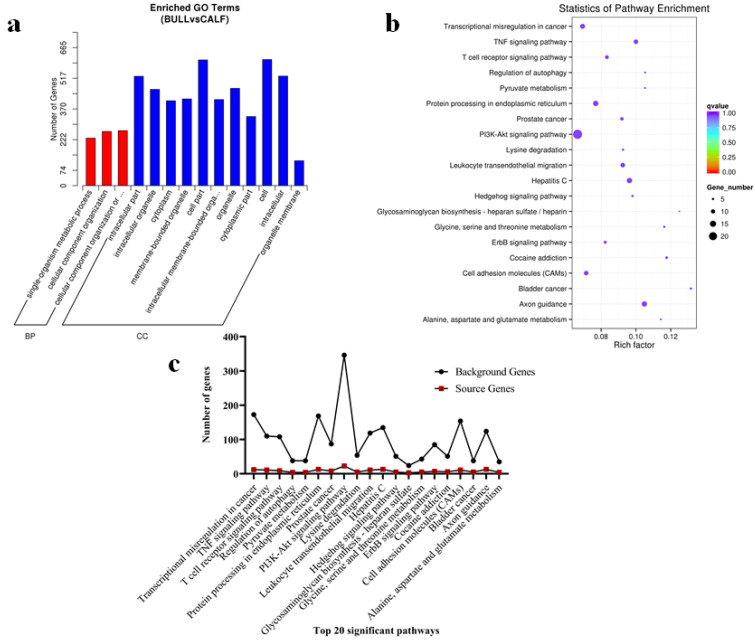
GO analysis and KEGG functional validation of DE miRNAs were highlighted. (**a**) The X-axis depicts the GO descriptions, while the Y-axis depicts the number of candidate genes identified in GO terms. The GO classifications are showing the two colored sections, as biological processes (red color) and cellular components (blue color). (**b**) Top-20 KEGG enriched pathways map of DE miRNAs linked genes in testicular samples. The vertical line shows the significant pathway names while horizontal line shows enrichment factors. (**c**) The top-20 signaling pathways represent the complete background genes and source genes.

**Figure 8 genes-13-02185-f008:**
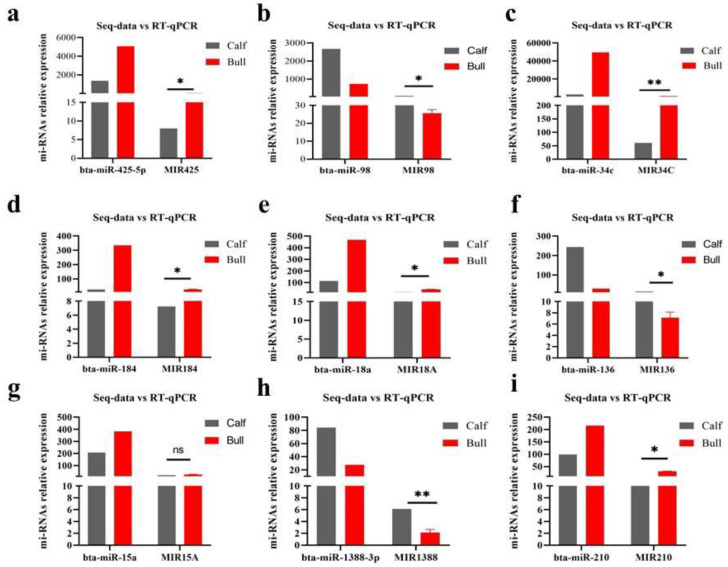
Validation of DE miRNAs via RT-qPCR was performed. (**a**–**i**) Shows the expression patterns of nine DE miRNAs (MIR425, MIR98, MIR34C, MIR184, MIR18A, MIR136, MIR15A, MIR1388, and MIR210) in bull and calf groups. The data were assessed by the 2^−ΔΔCt^ method with the U6 small nuclear RNA as an internal standard. The data are presented as the MEAN ± SEM; ** p* < 0.05, *** p* < 0.01.

**Figure 9 genes-13-02185-f009:**
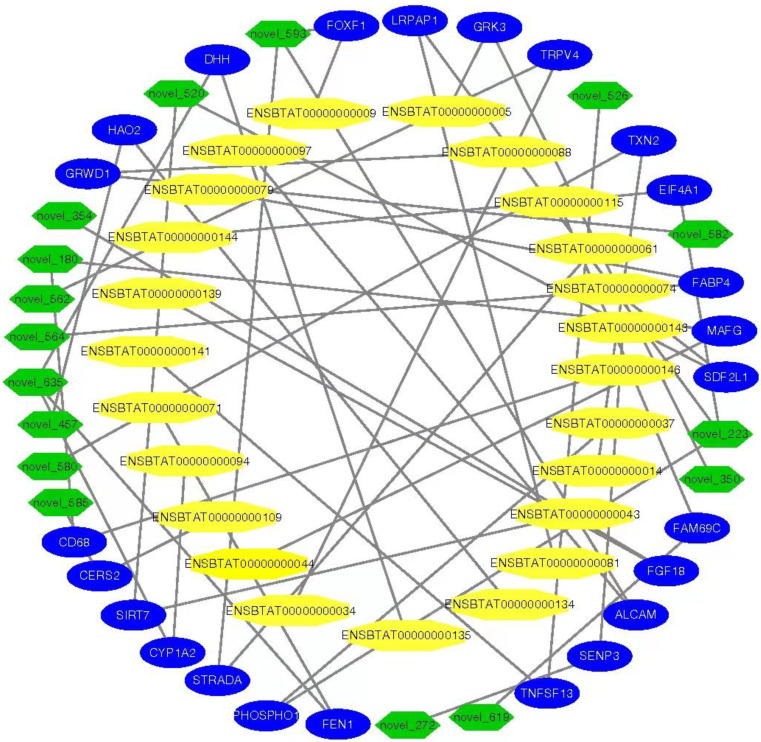
The interactive network of miRNAs, mRNAs and targeted genes were constructed between bull and calf testes. The green-colored nodes exhibit the miRNAs, blue highlights the link target genes, while the yellow color shows the mRNA.

**Table 1 genes-13-02185-t001:** The miRNAs forward primers were used in the RTq-PCR experiments.

Micro-RNA	Gen-Bank. No	Forward Primer	Reverse-Primer	Tm
bta-miR-425-5p	NR_030967.1	GCCGAGATGACACGATCACT	Universal Fix-Seq	55 °C
bta-miR-98	NR_031360.1	GCCGAGTGAGGTAGTAAGTT	Universal Fix-Seq	55 °C
bta-miR-34c	NR_030942.1	GCCGAG AGGCAGTGTAGTTA	Universal Fix-Seq	55 °C
bta-miR-184	NR_031179.1	GCCGAG TGGACGGAGAACTG	Universal Fix-Seq	55 °C
bta-miR-18a	NR_030892.1	GCCGAGTAAGGTGCATCTAG	Universal Fix-Seq	55 °C
bta-miR-136	NR_030865.1	GCCGAGACTCCATTTGTTTTG	Universal Fix-Seq	60 °C
bta-miR-15a	NR_030793.1	GCCGAGTAGCAGCACATAA	Universal Fix-Seq	65 °C
bta-miR-1388-3p	NR_036335.1	GCCGAG ATCTCAGGTTTGTUC	Universal Fix-Seq	65 °C
bta-miR-210	NR_049717.1	GCCGAGACTGTGCGTGTGACA	Universal Fix-Seq	65 °C

**Table 2 genes-13-02185-t002:** Raw reads sequencing data results and quality.

Sample	Reads	Bases	Error Rate	Q20	Q30	GC Content
Bull_1	23004713	1.150G	0.01%	98.57%	95.35%	48.78%
Bull_2	19243007	0.962G	0.01%	99.25%	97.13%	48.71%
Bull_3	21312453	1.066G	0.01%	98.48%	95.11%	48.32%
Calf_1	17862525	0.893G	0.01%	98.51%	95.28%	48.84%
Calf_2	17696661	0.885G	0.01%	99.35%	97.56%	48.79%
Calf_3	21796590	1.090G	0.01%	98.49%	95.07%	48.99%

**Table 3 genes-13-02185-t003:** Enriched GO terms and DEGs correlated with bull reproduction were identified.

Groups	Terms	DEGs No	GO_Accession	Genes ID
Calf vs. Bull	Sexual reproduction	14	0019953	*CFAP157*, *SEPT6*, *CCDC36*, *MEI1*, *FOLR2*, *BBS1*, *TYRO3*, *B4GALNT1*, *BCL6*, *BSP3*, *JAG2*, *PHC2*, *TGFBR1*, *CTDNEP1*
	Male gonad development	4	0008584	*NR5A1*, *TGFBR1*, *SF1*, *NKX3-1*
	Germ cell development	4	0007281	*SH2B1*, *CDKN1A*, *HYAL1*, *MYOG*
	Germ cell migration	1	0008354	*TGFBR1*
	Spermatid development	3	0007286	*MEI1*, *BSP3*, *CFAP157*
	Fusion of sperm to egg plasma membrane	1	0007342	*FOLR2*
	Sperm part	3	0097223	*TMEM190*, *SEPTIN6*, *TEKT3*

## Data Availability

All data generated or analyzed during this study are available from the corresponding authors upon reasonable request.

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
