# Peer review of "A Comprehensive Sequencing Analysis of Testis-Born miRNAs in Immature and Mature Indigenous Wandong Cattle (Bos taurus)"

_genes, 2022, doi:10.3390/genes13122185_

Round 1

Reviewer 1 Report

The article entitled "A Comprehensive Sequencing Analysis of Testis-born miRNAs in Immature and Mature Indigenous Wandong Cattle (Bos taurus)" described a method to understand the role of miRNAs in the development of testis and spermatogenesis. This article contains very detailed information regarding methods, which nowadays are very scarce. Also, the analysis and bioinformatics are very clear, leading to the reader's comfort. I only suggest improving the figures title's, once they are very small and difficult to read.

Author Response

Reply to Reviewer Comments for Manuscript ID: genes-1999853

“A Comprehensive Sequencing Analysis of Testis-born miRNAs in Immature
and Mature Indigenous Wandong Cattle (Bos taurus)”

Dear Editor and Reviewer!

We would like to commence by thanking the editor and the reviewers for their valuable time and constructive comments. We believe that the comments have been highly constructive and very useful to improve the quality of the revised manuscript. Thus, we have thoughtfully carried out revisions based on the comments and suggestions. Here, all of the changed texts have been red highlighted in our revised manuscript as shown in a word file entitled “revised manuscript”.

We hope that all these revisions in the revised manuscript will be sufficient to make our manuscript acceptable for publication in Genes MDPI. Please let us know if any further clarifications are necessary.

Sincerely,

Authors

Reviewer #1

Comments and Suggestions for Authors

The article entitled "A Comprehensive Sequencing Analysis of Testis-born miRNAs in Immature and Mature Indigenous Wandong Cattle (Bos taurus)" described a method to understand the role of miRNAs in the development of testis and spermatogenesis. This article contains very detailed information regarding methods, which nowadays are very scarce. Also, the analysis and bioinformatics are very clear, leading to the reader's comfort. I only suggest improving the figures title's, once they are very small and difficult to read.

Response: Thank you very much for the nice comments. We have thoroughly revised the figures legends and the changed typescript are red highlighted in our revised manuscript.

Reviewer 2 Report

The authors profiled and compared miRNAs in the testis between two small groups of cows (before mature and mature) to look into the potential function of miRNA in the bovine testicular development. The presentation of the manuscript is clear and precise. However, there are some minor grammar errors in the text; please check them out carefully. 

This manuscript provides a set of resources on the miRNA expression in the bovine testis and testicular development. 

Author Response

Reply to Reviewer Comments for Manuscript ID: genes-1999853

“A Comprehensive Sequencing Analysis of Testis-born miRNAs in Immature
and Mature Indigenous Wandong Cattle (Bos taurus)”

Dear Editor and Reviewer!

We would like to commence by thanking the editor and the reviewers for their valuable time and constructive comments. We believe that the comments have been highly constructive and very useful to improve the quality of the revised manuscript. Thus, we have thoughtfully carried out revisions based on the comments and suggestions. Here, all of the changed texts have been red highlighted in our revised manuscript as shown in a word file entitled “revised manuscript”.

We hope that all these revisions in the revised manuscript will be sufficient to make our manuscript acceptable for publication in Genes MDPI. Please let us know if any further clarifications are necessary.

Sincerely,

Authors

Reviewer # 2

Comments and Suggestions for Authors

The authors profiled and compared miRNAs in the testis between two small groups of cows (before mature and mature) to look into the potential function of miRNA in the bovine testicular development. The presentation of the manuscript is clear and precise. However, there are some minor grammar errors in the text; please check them out carefully. 

This manuscript provides a set of resources on the miRNA expression in the bovine testis and testicular development. 

Response: Thank you very much for the nice comments. The whole manuscript was thoughtfully carried out for English proofread by a native speaker. The change are red highlighted in the revised manuscript.
